# Laboratory experiments on the influence of stratification and a bottom sill on seiche damping

Karim Medjdoub[1], Imre M. Jánosi[2,3], and Miklós Vincze[1,4]

[1]von Kármán Laboratory of Environmental Flows; Eötvös Loránd University, Pázmány P. sétány 1/A, Budapest H-1117, Hungary
[2]Max Planck Institute for the Physics of Complex Systems, Nöthnitzer Str. 38, 01187 Dresden, Germany
[3]University of Public Service, Faculty of Water Sciences, Ludovika sqr. 1, H-1083 Budapest, Hungary
[4]MTA-ELTE Theoretical Physics Research Group, Pázmány P. sétány 1/A, Budapest H-1117, Hungary

**Correspondence:** Miklós Vincze (mvincze@general.elte.hu)

**Abstract.** The damping of water surface standing waves (seiche modes) and the associated excitation of baroclinic internal waves are studied experimentally in a quasi-two-layer laboratory setting with a topographic obstacle at the bottom, representing a seabed sill. We find that topography-induced baroclinic wave drag contributes markedly to seiche damping in such systems. Two major pathways of barotropic-baroclinic energy conversions were observed: the stronger one – involving short-wavelength internal modes of large amplitudes – may occur when the node of the surface seiche is situated above the close vicinity of the sill. The weaker, less significant other pathway is the excitation of long waves, internal seiches along the pycnocline that may resonate with the low frequency components of the decaying surface forcing.

## 1 Introduction

Energy conversion over bathymetric formations is a key component of the global ocean dynamics and mixing (Wunsch and Ferrari, 2004). The coupling between barotropic tidal waves at the sea surface and internal gravity waves facilitates heat and material exchange between the uppermost and deeper layers (Garrett, 2003; Lelong and Kunze, 2013; Morozov, 2018; Vic et al., 2019; Rippeth and Green, 2020; Stanev and Ricker, 2020). This largely interconnected dynamics is particularly pertinent in semi-enclosed basins, bays, and fjords with density profiles characterized by sharp gradients (Rattray Jr., 1960; Niiler, 1968; Bell Jr., 1975; Stigebrandt, 1980; Stigebrandt and Aure, 1989; Chapman and Giese, 1990; Münnich, 1996; Parsmar and Stigebrandt, 1997; Stigebrandt, 1999; Antenucci and Imberger, 2001, 2003; Inall et al., 2004; Cushman-Roisin et al., 2005; Johnsson et al., 2007; Boegman and Ivey, 2012; Park et al., 2016; Staalstrøm and Røed, 2016; Castillo et al., 2017; Roget et al., 2017; Stanev and Ricker, 2020; Xue et al., 2020).

The exchange between waves at the surface and at the pycnocline can be especially well studied in the situation where standing waves – seiches – develop at the water surface, known as seiches (Chapman and Giese, 2001; de Carvalho Bueno et al., 2020). Such oscillatory motions of a water body are typically initiated by temporally changing wind stress, especially pulse-like wind bursts, or storms. Strong wind shear gets balanced by a certain tilt of the water surface throughout the basin, but when the storm subsides, the restoring forces, predominantly gravity, yield a damped "sloshing" of the free surface, until it

settles in its equilibrium (horizontal) position. Tidal barotropic waves, even tsunamis and other seismic disturbances are known to generate large inflows into coastal harbors and may also yield strong seiche activity (Chapman and Giese, 2001).

The energy dissipation rate (or decay rate) of surface seiches in natural enclosed water bodies, e.g. lakes or fjords is determined by the physical properties of the fluid body (e.g. its stratification profile) and the geometry of the basin. In the presence of bottom topography, surface gravity waves generate internal waves, a process also referred to as barotropic to baroclinic energy conversion, or – in the specific case of tidal surface waves – tidal conversion. The first demonstration of such a wave excitation mechanism over a tilted bottom topography dates back to the work of Rattray Jr. (1960) in the case of a two-layer

stratification. Niiler (1968) explained the observed current oscillation at the Florida Straits by interaction of barotropic tides and the continental slope generating baroclinic tidal waves, periodically modulating the flow. Although these pioneering studies did not consider the dissipative effect of the excited baroclinic modes on the surface waves, it is obvious that freely propagating internal waves use up a significant fraction of the energy stored in the surface oscillations, and hence speed up the decay of the latter.

There are, however, substantial differences between the seiche modes excited in enclosed and semi-enclosed basins. In the former configurations the boundary conditions "select" such standing waves that have antinodes at all boundaries (at least in the idealized case of vertical sidewalls), whereas, in the semi-enclosed systems the open boundary (e.g. the mouth of the bay) facilitates maximum horizontal flux, leading to the formation of a wave node at the surface. Obviously, the internal waves generated by the waves over topographic obstacles may also exhibit different behaviour in the two settings. In the semi-infinite

domain travelling internal waves are excited which "radiate" away the kinetic energy of the surface waves when the topographic obstacles at the bottom reach up to the pycnocline (Davies et al., 2009). If, however, the basin is closed, resonant mode selection can also occur as internal seiches are generated at the pycnocline.

The effect of damping on the frequencies of the seiche modes was addressed by the paper of Cushman-Roisin et al. (2005) who concluded that in enclosed water bodies the oscillation periods of surface seiche modes do not depend on the stratification

significantly. However, in semi-enclosed systems the surface seiche period was found to be more sensitive to the stratification. This effect of internal wave excitation on the decay of surface seiches was further investigated by Wynne et al. (2019). When the characteristic (e.g. diurnal) frequency of the wind forcing matches that of a seiche eigenmode, a resonant amplification of the latter may occur. However, as the seasonal changes of the freshwater inflow (e.g. from glacier runoff) influence the depth of the pycnocline, the natural frequencies of the internal waves also change, and – as the resonant frequencies shift – so does

the decay coefficient of the dominant surface seiche mode, as shown by Antenucci and Imberger (2001, 2003).

The basin geometry of certain fjords, e.g. the Gullmar fjord of Sweden is especially interesting, as their bottom topography involves a sill reaching up to the pycnocline between the saline seawater and the upper freshwater layer. Parsmar and Stige-brandt (1997) investigated the damping of tidally excited surface seiches in the Gullmar fjord and concluded that the drag caused by baroclinic internal wave excitation at the sill contributes far more to the damping of the seiche than bottom friction

or other phenomena.

Earlier work in our laboratory (Vincze and Bozóki, 2017) investigated interfacial internal wave generation experimentally in a quasi-two-layer wave-tank with a bottom obstacle at its center, reaching up to the pycnocline, motivated by the fjord geometry.

We found that shear instability developing at the tip of the obstacle and the emerging billows press down the pycnocline at the lee side of the obstacle (with respect to the direction of the oscillating horizontal flow in the upper layer) and generate the observed internal waves. However, the energetics of the damping of surface seiches was not addressed there.

Internal wave excitation by surface seiche modes has also been investigated in laboratory experiments by Boegman and Ivey (2012). They modeled resonant barotropic-baroclinic energy conversion of a periodically forced shallow water system in a rectangular tank without topographic obstacles in order to quantify the energy flux pathways between the applied forcing and the internal wave field. A similar setting (i.e. a rectangular tank with a flat bottom) has also been studied recently by Xue et al. (2020), with a special focus on internal wave excitation and seiche damping. Similar laboratory models can provide illuminating results for the interpretation of direct oceanic observations, because they make possible to focus on specific phenomena that are inevitably loaded with disturbing effects in natural environments.

Here we report on laboratory experiments in a narrow water tank, filled up with quasi two-layer salinity stratified water, with a vertical obstacle installed in the middle of the domain in the bottom layer. We analyze the damping of various surface and internal wave modes that develop after pulse-like initial seiche excitation at the water surface. To the best of our knowledge, these are the first experiments in the literature on surface seiche damping in the presence of a bottom sill yielding considerable baroclinic wave drag.

The paper is organized as follows. Section 2 describes the experimental set-up and the applied data acquisition methods. In Section 3 we present our results on the relationship between surface seiche damping and the physical and geometrical properties of the experimental settings. We then briefly discuss the implications of our findings to actual fjords in Section 4 and summarize our findings in Section 5.

## 2   Set-up and methods

Our experiments were conducted in a transparent rectangular acrylic (Plexiglas) tank of length $L = 80$ cm and width $w = 5$ cm, filled up with quasi-two-layer stratified water, as shown in Fig. 1a. The bottom layer of thickness $H_2 = 4$ cm consisted of saline water colored with blue food dye, whereas the upper layer of thickness $H_1$ was freshwater of the same temperature, colored red. To inhibit mixing at the interface (i.e. diapycnal mixing) during the filling-up procedure the freshwater of the upper layer was entering the surface slowly, through a sponge.

With each prepared stratification profile two configurations were studied. One had an upside-down U-shaped small plastic obstacle (Fig. 1a) that was placed in the middle of the tank to the bottom. The obstacle's height $h$ matched the thickness of the bottom (saline) layer: $h = H_2 = 4$ cm. In the other series of experiments the obstacle was removed from the tank (slowly, in the vertical direction, to minimize turbulent mixing) and the dynamics was studied in the absence of the topographic obstacle. The experimental parameters and the list of other notations used throughout the paper are given in Table 1.

Each experiment series consisted of five runs with and five runs without the obstacle. A run was initiated as follows. A suspended wave maker consisting of a metallic rod and six rectangular rubber foam "bumpers" (Fig. 1b) was placed above the water surface. The bumpers could be placed in various configurations along the rod to facilitate the excitation of surface seiche

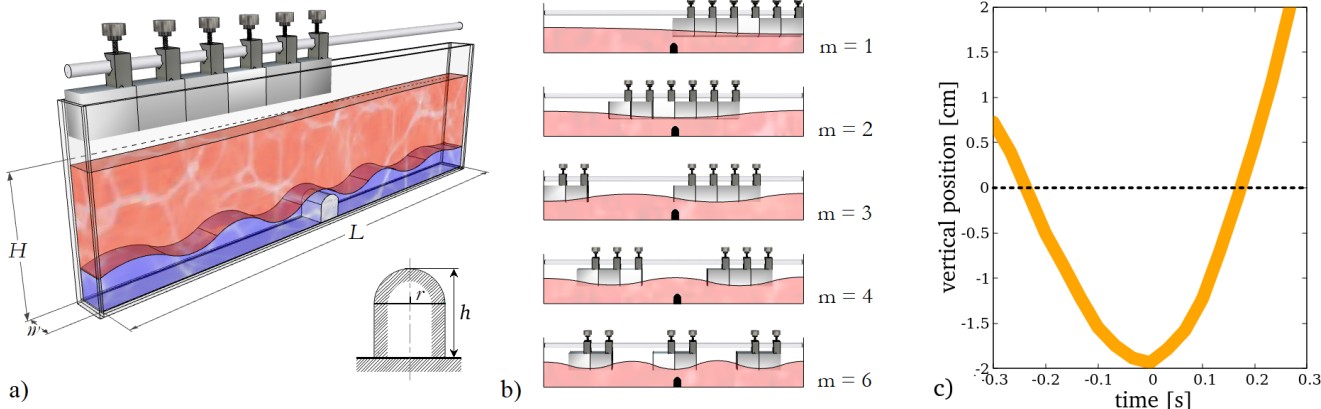

**Figure 1.** (a) The schematics of the set-up. The geometrical parameters of the tank are $L = 80$ cm, $w = 5$ cm, and $H_2 = h = 4$ cm. The shape of the obstacle is sketched in the bottom right corner ($h = 4$ cm, $r = 1.75$ cm). (b) The configurations of the six rubber foam bumpers of the wave maker for the excitation of the various surface seiche modes and the sketch of the corresponding waveform. (c) The vertical motion of the bottom of the rubber foam bumpers at the initiation. The position is given with respect to the unperturbed water surface. The width of the line represents the error of reproducibility.

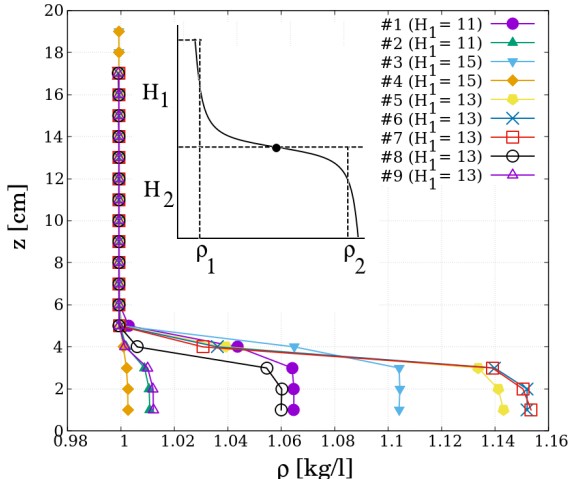

**Figure 2.** Density $\rho$ as a function of vertical position $z$ from the bottom, for all experiments series (cf. Table 2). The sketch in the inset demonstrates the meaning of the parameters of the two-layer approximation.

(standing wave) modes, as sketched in Fig. 1b. The standing wave pattern with a dominant mode of $m = 5$ – where $m$ denotes the number of nodes of the standing waveform – was found to be hard to excite with this device. But since we did not intend

**Table 1.** List of parameters and notations.

| notation | meaning | value | unit |
|---|---|---|---|
| $L$ | length of the tank | 80 | cm |
| $w$ | width of the tank | 5 | cm |
| $H$ | total fluid depth | 15;17;19 | cm |
| $H_1$ | upper layer depth | 11;13;15 | cm |
| $H_2$ | bottom layer depth | 4 | cm |
| $h(=H_2)$ | obstacle height | 4 | cm |
| $H_r(=H_1 H_2/H)$ | reduced height | 2.9–3.2 | cm |
| $\rho_1$ | upper layer density | 0.998 | kg/l |
| $\rho_2$ | bottom layer average density | 1.001–1.114 | kg/l |
| $\Delta\rho(=\rho_2-\rho_1)$ | density contrast | 0.003–0.116 | kg/l |
| $k$ | wavenumber | | rad/cm |
| $m$ | dominant surface mode number | 1;2;3;4;6 | – |
| $\omega$ | angular frequency | | rad/s |
| $f(=\omega/(2\pi))$ | frequency | | Hz |
| $\lambda(=2\pi/k)$ | wavelength | | cm |
| $\eta$ | surface displacement | | cm |
| $\chi$ | pycnocline displacement | | cm |
| $A_i$ | amplitude of surface wave component $i$ | | cm |
| $C_i$ | decay coefficient of surface wave component $i$ | | 1/s |
| $\varphi_i$ | phase of surface wave component $i$ | | rad |

**Table 2.** Geometrical and physical characteristics of the experiment series.

| Experiment series | #1 | #2 | #3 | #4 | #5 | #6 | #7 | #8 | #9 |
|---|---|---|---|---|---|---|---|---|---|
| $H_1$ (cm) | 11 | 11 | 15 | 15 | 13 | 13 | 13 | 13 | 13 |
| $\rho_2$ (kg/l) | 1.0592 | 1.0076 | 1.0872 | 1.0019 | 1.1114 | 1.1136 | 1.1114 | 1.0437 | 1.0084 |

to study perfectly pure eigenmodes, and it was not crucial for our analysis to excite all possible dominant modes, we restricted our evaluation to dominant surface seiche modes $m = 1, 2, 3, 4,$ and 6. In the beginning of each run, the suspended rod was pushed down (only once) to the water surface, reaching a maximum depth of $1.65 \pm 0.25$ cm in a standardized manner and was then instantaneously pulled up, out of the water. The vertical motion of the bottom of the bumper at initiation is shown in Fig. 1c, as acquired from video tracking. Apparently, the characteristic timescale of the impact, i.e. the duration that the


bumper spent below the unperturbed water height (indicated with dashed line) was approximately $(0.4 \pm 0.06)$ s, i.e. shorter than the basin-crossing timescale of a surface perturbation.

The prepared density profiles of the experiment series are presented in Fig. 2, as measured by a conductivity probe. The physical parameters of the two-layer approximation, namely layer thicknesses $H_1$ and $H_2$ and the respective densities $\rho_1$ and $\rho_2$ were obtained as follows. $H_1$ and $H_2$ were measured directly (with a ruler) based on their color (and indeed, the intended thickness $H_2 = h = 4$ cm matching the obstacle height could be achieved within $\pm 0.25$ cm in all cases). The characteristic density $\rho_2$ of the saline layer was taken as an average of the measured density profile in the $z \leq H_2$ domain, as sketched in

the inset of Fig. 2. The relevant adjustable parameters of the stratification are summarized in Table 2 for all nine experiment settings.

     The damping of the initiated surface sloshing and the internal dynamics were recorded by an HD camera (at frame rate 30 fps and frame size 1080 px $\times$ 1920 px) facing the long sidewall of the tank perpendicularly, close to the middle of the tank. Afterwards, the open source correlation based feature tracking software Tracker (https://physlets.org/tracker/) was used

to acquire the time series of surface and pycnocline motion.

## 3    Results

### 3.1    Qualitative description

The flow dynamics in the set-up is driven by the decaying surface waves initiated by the aforementioned standardized instantaneous push of the wave maker. In the qualitative sense, the decay of the resulting surface standing waves is not affected by the

presence or absence of the bottom obstacle (although the actual values of the decay rates are substantially different in the two configurations as will be discussed later). However, the dynamics at the pycnocline can be vastly different in the two cases. If the obstacle is installed, velocity shear can develop between the bottom and top layers that yields the excitation of baroclinic (internal) wave modes along the pycnocline (Vincze and Bozóki, 2017). In the control runs – i.e. without obstacle – the vertical displacement of the surface and the pycnocline are found to be co-aligned at each time instant, as if in a homogeneous fluid

(barotropic wave modes).

     Yet, even if the obstacle is present, large amplitude internal waves do not necessarily get excited, as demonstrated in Figs. 3 and 4. Fig. 3 shows snapshots from two experiments, characterized by the leading mode of the surface seiche that is $m = 1$ (base mode) and $m = 2$ in the top and bottom images, respectively. One can notice that in the former case, where a node of the surface wave form is located at the center of the tank, indicating purely horizontal flow right above the obstacle, slowly

propagating internal waves of short wavelength appear "radiating away" from the obstacle with fairly large amplitudes. In the latter experiment, however, where the surface displacement has an antinode – and hence vertical oscillation of the fluid parcels – at the obstacle location, no progressive internal waves are visible.

     The corresponding space-time plots showing the displacement of the surface $\eta(x,t)$ and of the pycnocline $\chi(x,t)$ with respect to their initial levels are presented in Figs. 4a and 4b, respectively, as obtained via video analysis. (Note, that the

color scales are different in the four panels.) The dashed vertical lines in Fig. 4a represent the locations of the nodes of the

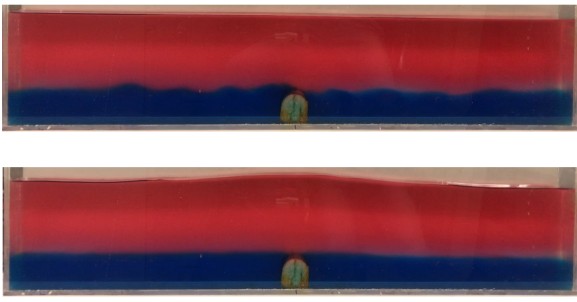

**Figure 3.** (a) Snapshots from experiments with $m = 1$ (top) and $m = 2$ (bottom) surface seiche modes (configuration #3, cf. Table 2).

dominant mode ($m = 1$ in the left and $m = 2$ in the right panels). After the initiation of the surface seiche at $t = 0$, propagating baroclinic waves dominate the picture in the $m = 1$ run (Fig. 4b left); later the waves get reflected at the sidewall and thus evolve into internal standing wave-like interference patterns. In the $m = 2$ case, the large-scale decaying barotropic oscillation characterizes the entire domain with hardly any noticeable slowly propagating structures (Fig. 4b right). This duality of internal

wave excitation is apparent throughout the higher seiche modes as well: antisymmetric surface wave forms associated with odd values of $m$ (see Fig. 1b) tend to excite larger baroclinic wave activity via the obstacle placed in the geometrical center than the symmetric waves (even values of $m$).

### 3.2 Surface waves

Vertical displacement time series of the water surface $\eta(x,t)$ and the pycnocline $\chi(x,t)$ were logged at the vicinity of the

sidewall (i.e. at $x \approx 0$ or $L$) in each experiment to ensure that all standing wave modes have antinode at the measurement location. Note, that due to the boundary conditions only here it is guaranteed that all standing wave components exhibit maximum amplitude. Examples of such records are shown in the panels of Fig. 5a and b for the $\eta$ and $\chi$ signals (red and blue curves, respectively) from runs with dominant modes $m = 1, 2$, and 4 (from the left).

The surface time series acquired at location $x = 0$ can be approximated as a sum of decaying sinusoidal oscillations in the

following form:

$$\eta(0,t) \approx \sum_{i=1}^{N} A_i \exp(-C_i t) \sin(\omega_i t + \varphi_i) \ , \tag{1}$$

where $\omega_i$, $\varphi_i$ and $A_i$ denote the frequency, phase shift and initial amplitude of the $i$-th component, respectively, whereas $C_i(\omega_i)$ is the decay coefficient of the given mode. Eq. (1) was fitted to the time series with $N = 2$ as the limit. The standard deviations of the residuals indicated that such two-term sums were sufficient to account for over 90% of the observed displacements in all

cases and in many cases even the single-mode fit ($N = 1$) was enough to reach the same precision. It is to be noted here that by using Eq. (1), the frequency-dependant coefficients of decay $C_i(\omega_i)$ could theoretically be extracted from the "sloshing" of the water surface with any arbitrary initial condition. We chose our initiation method with the different bumper configurations

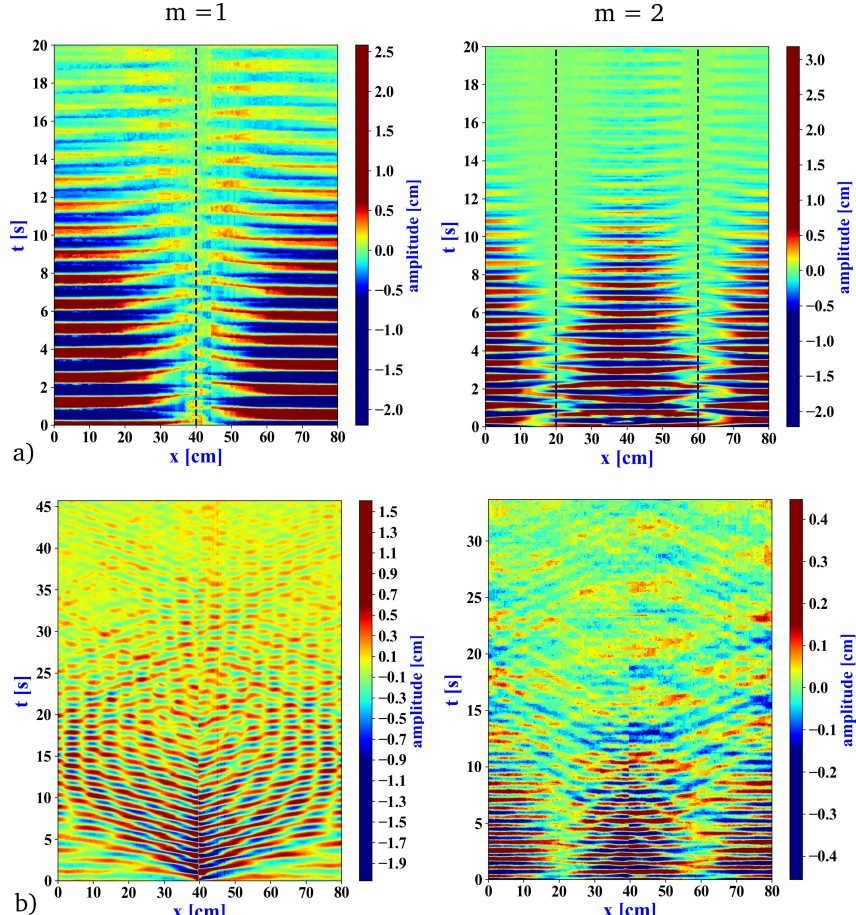

**Figure 4.** Space-time plots of the two experiments of Fig. 3, showing the vertical dispacement of the surface $\eta(x,t)$ (a) and the pycnocline $\chi(x,t)$ (b) in the whole tank for dominant surface wave modes $m = 1$ (left) and $m = 2$ right (obstacle location is at $x = 40$ cm). Vertical dashed lines in the top panels denote the expected locations of nodes.

(shown in Fig. 1b) representing "quasi pure" modes only for the practical reason of making the regressions to the time series as simple as reasonably achievable.

The Fourier amplitude spectra $A_\eta(f)$ and $A_\chi(f)$ of the $\eta(t)$ and $\chi(t)$ signals are presented in Fig. 5c with a color coding identical to the corresponding time series themselves. The frequencies of the largest spectral peaks of $A_\eta(f)$ from each experiment were found to be in good agreement (i.e. within 5%) with the linear dispersion relation of homogeneous (non-stratified) surface gravity waves, which reads as

$$\omega = \sqrt{gk\tanh(kH)} \ , \tag{2}$$

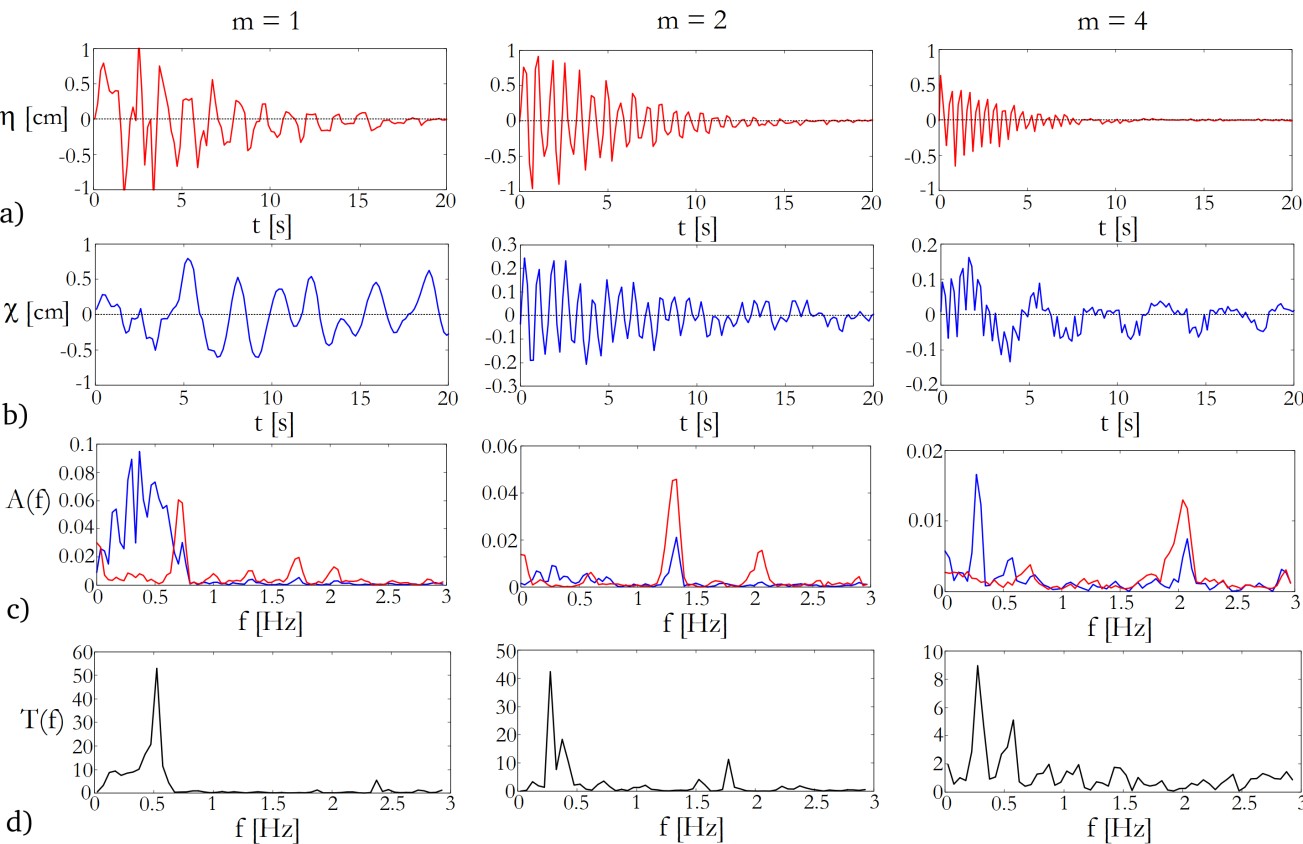

**Figure 5.** Time series and spectra from three exemplary runs (configuration #1, cf. Table 2) of dominant modes $m = 1$, $m = 2$, and $m = 4$ from the left. The panels of row a) show the water surface displacement as a function of time $t$ from the initiation, as tracked in the close vicinity of the sidewall. The displacement of the pycnocline $\chi$ at the sidewall is presented in row b) (note, that here, unlike in the $\eta(t)$ plots, the vertical range is different in all three panels). The Fourier spectra of the $\eta(t)$ (red) and $\chi(t)$ (blue) records are visible in row c), whereas the panels in d) show the transfer function $T(f) = A_\chi(f)/A_\eta(f)$, i.e. the ratio of the spectra of the corresponding two spectra above.

where $H$ is the total water depth, $g$ is the gravitational acceleration, and the wave number $k$ is to be taken at $k(m) = \pi m/L$ for the dominant standing wave mode excited in the given run (cf. Fig. 1). The match is demonstrated in Fig. 6. In terms of the $\omega(m)$ relation, no systematic bias was found from this simple (linear, undamped) theoretical formula regardless of whether the experiment included the obstacle or not, or even whether the water was stratified or homogeneous.

### 3.3 Source-filter dynamics

The surface waves exhibit faster damping than their internal counterparts and the dominant frequency components of the two may also largely differ. It is clearly visible from Fig. 5 that the "internal" $\chi(t)$ signals (panels in row b) possess more

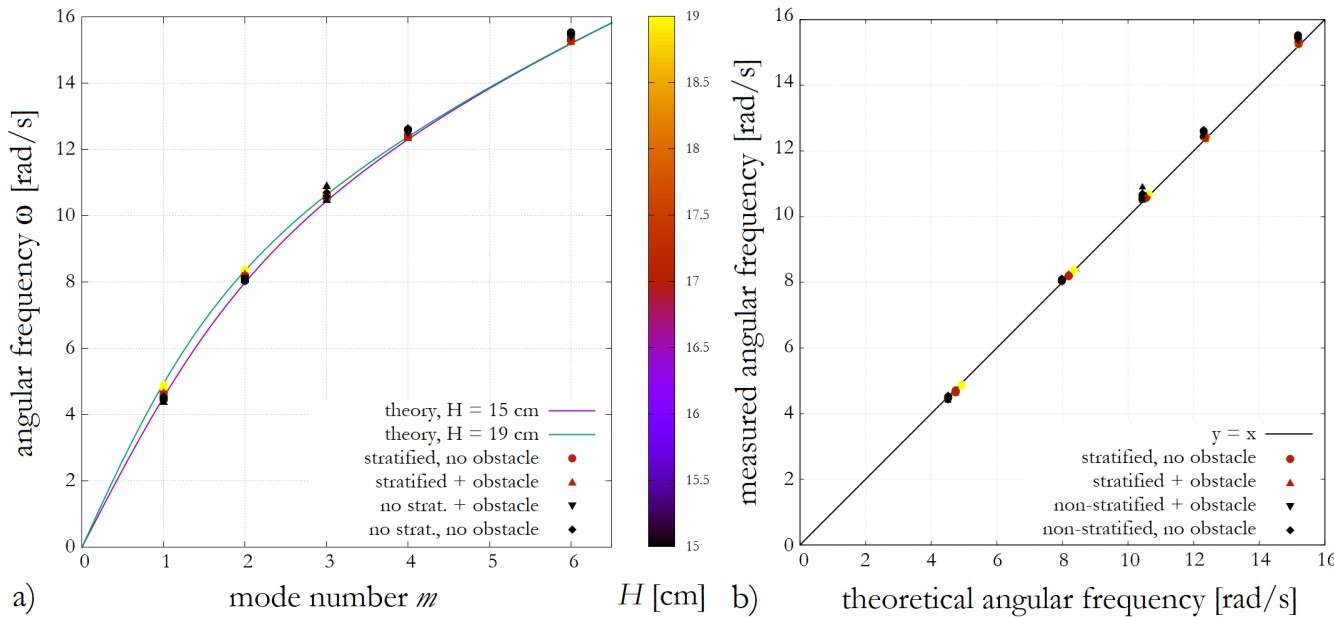

**Figure 6.** (a) The dispersion relation of water surface waves Eq. (2) expressed as a function of dominant mode index $m$. The solid curves represent the theoretical relationship for the shallowest ($H = 15$ cm) and deepest ($H = 19$ cm) configurations, and the data points indicate the measured values of angular frequency $\omega$. The coloring is based on the total water height $H$. (b) Contrasting the calculated angular frequency $\omega$ from Eq. (2) to its measured value. The various experiment configurations are indicated by the shapes of the symbols in both panels (see legend).

pronounced low-frequency variability than the $\eta(t)$ oscillations (row a), even in the $m = 2$ and $4$ modes, where the classic, topography-induced baroclinic wave generation is practically inhibited – due to the lack of considerable velocity shear at the obstacle – as discussed earlier.

To understand this mechanism it is important to note that even if the surface seiche was a perfectly "monochromatic", single-frequency source signal, its exponential decay would still unavoidably introduce nonzero amplitudes into the low-frequency range of its spectrum (see, e.g. French (1971)), making it suitable for the excitation of slow internal oscillations. The resulting signal at the pycnocline can thus be understood as the outcome of a resonance-like amplification of certain characteristic frequency bands of the surface source signal.

The transfer function $T(f)$ of this frequency "filtering", a widely used tool, e.g. in acoustics (Stevens, 2001), can be defined as $T(f) = A_\chi(f)/A_\eta(f)$. Such empirical transfer functions for the experiments of Fig. 5 are shown in the panels of row d). In all cases, the maxima of $T(f)$ appear well below the fundamental frequencies $f_i = \omega_i/(2\pi)$ of the source signal Eq. (1), consistently with the rule of thumb that interfacial internal propagation is typically around $\sqrt{\Delta\rho/\rho_0}$ times slower than that of surface waves. The question arises of what kind of process determines the observed low-frequency amplification bands of

$T(f)$.

In the two-layer approximation, the dispersion relation of small amplitude interfacial internal waves (Massel, 2015) takes the form

$$\omega(k) = \sqrt{\frac{gk\Delta\rho}{\rho_1 \coth(H_1 k) + \rho_2 \coth(H_2 k)}} \ , \tag{3}$$

where $k$ denotes the wavenumber (in this case, along the pycnocline), $H_1$ and $H_2$ are the thicknesses and $\rho_1$ and $\rho_2$ are the densities of the top and bottom fluid layers, respectively (cf. Table 2), and $\Delta\rho = \rho_2 - \rho_1$ is their difference.

A straightforward way to contrast the theoretical dispersion relation with the observed wave propagation is taking the two-dimensional Fourier transform of space-time plots like the ones shown in Fig. 4. Then, the time axis is mapped onto the frequency domain $\omega$ and the spatial axis is transformed to the wavenumber ($k$) space. Marked spectral amplitudes are expected to occur along the $\pm\omega(k)$ dispersion relation curves. Examples of such spectra are presented in Fig. 7 for the cases of dominant surface seiche modes $m = 4$ and $m = 6$ in experiment series #3 (see Table 2) for the surface (panels a) and the pycnocline (panels b). In the surface spectra of panels a) the "quasi pure" initiation yielded sharp peaks (cf. Fig. 5) corresponding to the dominant modes (white circles) that indeed lie along the surface dispersion relation Eq. (2), plotted with black lines (cf. Fig. 6a). The spectra of the pycnocline motion exhibit much more disperse responses, nevertheless we also find that the formula of Eq. (3) shown again with black lines match the observed distribution of large amplitudes in the low wave-number domain. (Towards higher values of $k$, three-layer corrections may be necessary as shown e.g. in Vincze and Bozóki (2017).)

Combining the transfer functions $T(f)$ of all five experimental runs (corresponding to different dominant $m$ values) in a given stratification setting, one can define a cumulative transfer function $\overline{T}(f)$ by assigning the largest value of $T(f)$ obtained throughout all five runs to each frequency $f$, as represented by the gray shaded area of the combined spectrum in Fig. 8a. The dispersion relation Eq. (3) enables us to transform the $T(f)$ functions to the wave-number domain, as shown in Fig. 8b for four selected stratification settings. Here the frequency $f = \omega/(2\pi)$ is given on the horizontal axis and the wave-number on the vertical one is expressed in the nondimensional units of $L/\lambda = Lk/(2\pi)$, with $\lambda$ being the internal wavelength and $L$ the length of the tank. (This is a rescaled inverse of the positive branches of the solid black curves in Fig. 7b.) The curves therefore represent the inverse of the dispersion relation Eq. (3), and their color scale marks the corresponding normalized cumulative transfer function $\overline{\overline{T}}(f)$ (the normalization was carried out by setting the maximum of the $T(f)$ function as unity for each stratification).

By reorganizing the cumulative transfer functions this way, one can notice that the amplification peaks (high values of $\overline{T}$, colored yellow) tend to appear in the close vicinity of integer values of the nondimensional wave number, e.g. at $L/\lambda = 1; 2; 3; 5$ and $8$, as marked by dotted horizontal lines in Fig. 8b. These low frequency resonance peaks thus seem to be related to standing wave generation at the pycnocline – internal seiche modes – whose wavelengths can fit integer times into the tank. Despite having much lower oscillation frequencies than the main peak of the "source" signal, these modes can still get excited. It is due to the fact that the damping of the surface seiche excites non-negligible amplitudes in the low-frequency spectral domain with which the internal standing waves can resonate. It is to be emphasized that combining the $L/\lambda = n/2$ (where $n = 1, 2, \dots$) criterion with the dispersion relation Eq. (3) provides the same information as calculating the corresponding internal seiche frequencies directly from the geometrical parameters of the setting.

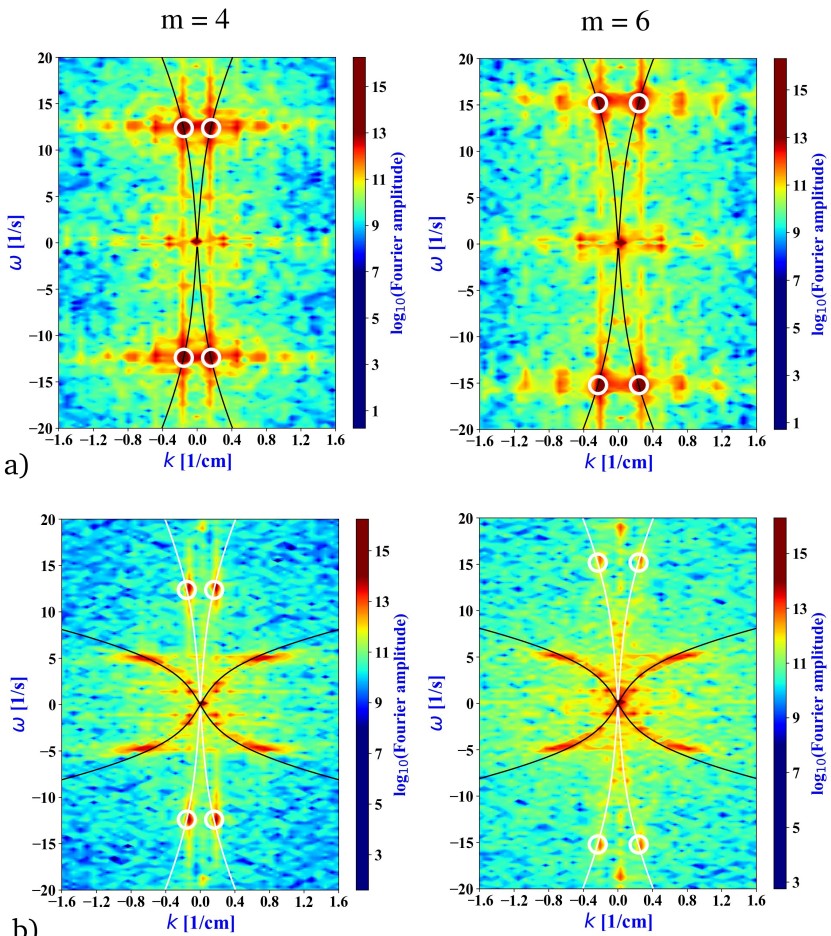

**Figure 7.** Two-dimensional Fourier transforms of surface displacement $\eta(x,t)$ (a) and of the pycnocline displacement $\chi(x,t)$ (b) in the tank for dominant surface seiche mode $m = 4$ (left) and $m = 6$ (right), experimental configuration #3. The white circles mark the dominant surface wave peak, whereas the black solid lines represent the respective dispersion relations Eqs. (2) and (3).

### 3.4 Topographic energy conversion

The damping of surface waves in a stratified system is caused by the combination of factors such as the friction with the basin boundaries, diapycnal mixing, and barotropic-to-baroclinic energy conversion due to topographic effects. To separate the contribution of the latter we conducted control experiments with each studied stratification profile where the obstacle was removed from the tank. We then compared the observed decay rates with the ones acquired with the obstacle in place.

The scatter plot in Fig. 9a contrasts the decay coefficient of the dominant surface seiche mode $C_o$ – from fitting formula (1) – in each experiment with obstacle (hence the index $o$), with the corresponding no-obstacle value $C_{no}$. It is clearly visible that

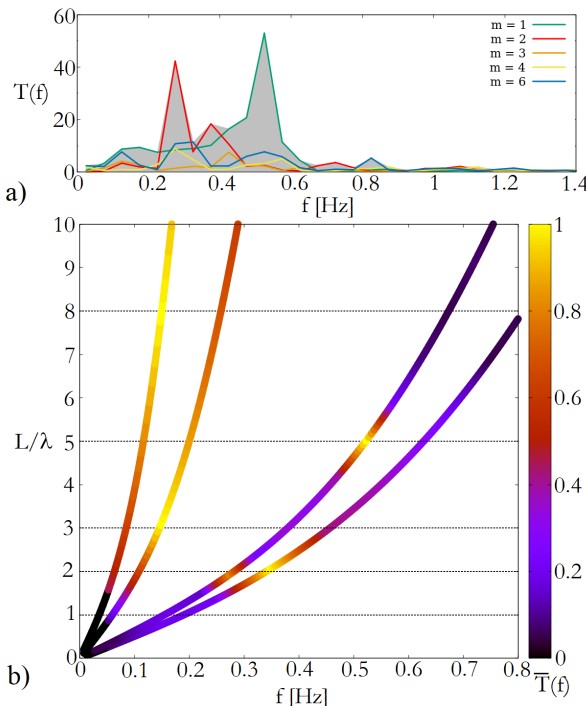

**Figure 8.** (a) The transfer functions $T(f)$ of all dominant excitation modes $m$ in experiment series #1, and their cumulative transfer function (grey shaded area). (b) The internal wave (inverse) dispersion relations as a function of frequency $f$ for four exemplary stratification settings (representing series #1, #2, #3, and #4). The coloring marks the normalized cumulative transfer functions $\overline{T}(f)$ as acquired from the corresponding experiment series.

$C_o > C_{\mathrm{no}}$ for almost all cases, but no systematic relationship could be found with the density ratio of the two layers (indicated by the coloring), or total water height $H$ (marked by the different symbol shapes, see legend).

There is, however, a marked difference between the two configurations in terms of the $m$-dependence of the decay coef-
ficients, as shown in Fig. 9b, where the $C_o/C_{\mathrm{no}}$ ratios are plotted for each experiment pair as a function of mode index $m$. Apparently, the relative increase of barotropic energy dissipation (damping) compared to the no-obstacle setting is the largest – up to a factor of 2.1 – at the odd wave-numbers $m = 1$ and 3 (note that the $m = 5$ mode could not be excited due to technical reasons, as mentioned earlier). This is in concert with the qualitative description provided in subsection 3.1, where we demonstrated that large amplitude internal waves having the same frequency as the surface wave are excited when the obstacle
is situated below a surface seiche node.

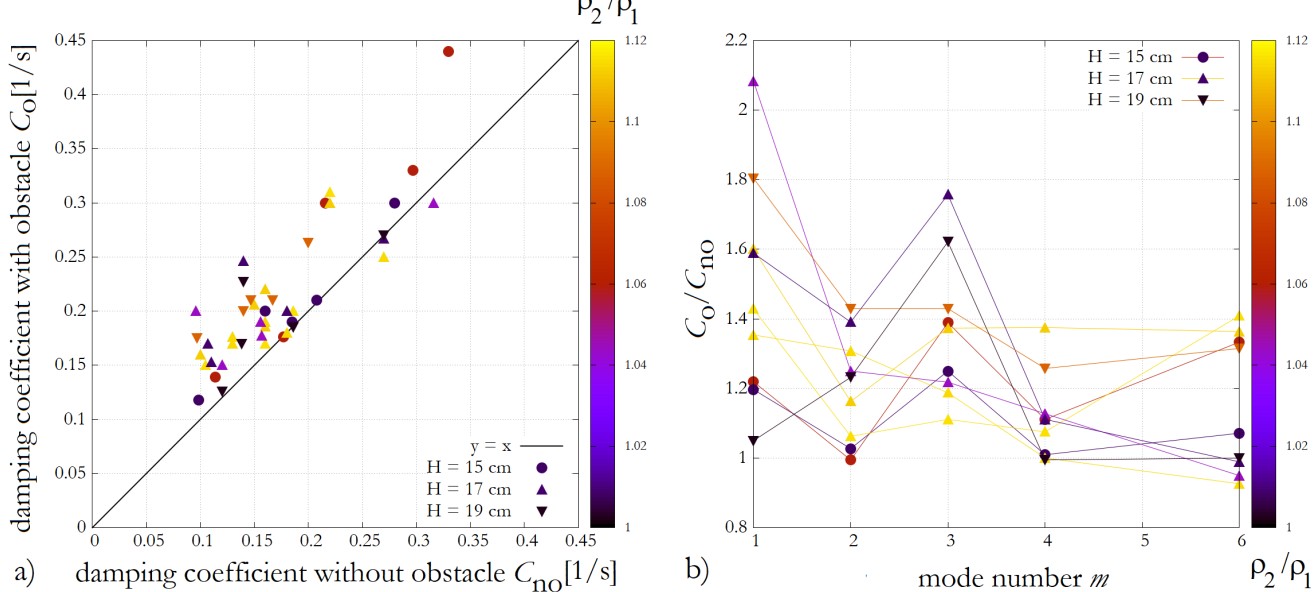

**Figure 9.** (a) Comparison of dominant mode surface seiche decay coefficients in paired experiments without (horizontal axis) and with (vertical axis) bottom obstacle. The solid line represents $y = x$. (b) The ratio of the decay coefficients $C_o/C_{no}$ of paired experiments with and without obstacle as a function of dominant surface seiche mode index $m$. Symbol shapes and coloring mark the total water depth and the density ratio of the two layers, respectively, in both panels (see legend).

## 4   Discussion

In what follows, we discuss the implications and applicability of our findings to actual sill fjords in nature, e.g. the Gullmar fjord of Sweden, whose internal wave excitation dynamics served as motivation for our work. For this particular fjord Parsmar and Stigebrandt (1997) applied an analytical two-layer model for a simplified rectangular geometry and – despite its simplicity

– have found a strikingly good agreement with field data when predicting the surface seiche decay coefficients $C$ using only the approximate geometrical dimensions of the fjord, and the densities and thicknesses of the layers.

Their formula (based on an earlier work, Stigebrandt (1976)) as applied to our experimental geometry, takes the form

$$C = \frac{8H_2\omega^2 L}{\pi^2 gHH_1} c_0, \tag{4}$$

where, as before, $\omega$ denotes the forcing frequency. The approximations used for eq. (4) – besides the ones mentioned above

– include linearity and the shallow-layer assumption, which manifests in the factor $c_0 = \sqrt{g\Delta\rho H_1 H_2/(\rho H)}$, i.e. the phase velocity of the interfacial internal waves in the (non-dispersive) long-wave limit. This limit is encountered when $kH_r \ll 1$ holds, where $k$ denotes the (horizontal) wavenumber and $H_r \equiv H_1 H_2/H$ is referred to as the "reduced height", a characteristic vertical scale of the system. Substituting the parameters of the Gullmar fjord or those of the Oslofjord (studied in Stigebrandt

(1976)) one indeed gets $kH_r \approx 0.02$ and $0.002$, respectively, due to the fact that the forcing frequency $\omega$ is on the order of $\sim 10^{-4}$ rad/s in both cases, yielding small values for $k\ (= \omega/c_0)$, well within the long-wave regime. In our laboratory setting, however, the internal waves are manifestly dispersive (cf. Figs. 7 and 8) and are found in the $H_r k > 1$ range even for the slowest excitation investigated. Hence here the full dispersion relation (3) has to be considered; (4) would significantly overestimate the decay coefficients.

The question arises of how relevant these experimental findings can then be for the better understanding of actual fjord systems. For instance, assuming an identical stratification profile to that of the Gullmar fjord, but a much shorter external (surface) seiche period of 100s, from the inverse of (3) we get $H_r k \approx 3.5$. Seiche periods of this timescale are not uncommon: the Norwegian Framfjorden for example exhibited surface standing waves with amplitudes reaching up to 1.5 m and with periods of $67-100$s triggered by the giant 2011 Tohoku earthquake, whose epicenter was located in the distant Japan (Bondevik et al., 2013). Our finding that the surface seiche-induced internal waves can be largely dispersive indicates that in such situations (short-period oscillatory forcing) a modified version of eq. (4) is to be applied, in which $c_0$ is replaced with the phase velocity $c = \omega/k(\omega)$, where $k(\omega)$ is the inverse function of (3) for a given forcing frequency $\omega$.

Another related observation is that the propagating interfacial internal waves appear to follow the dispersion relation derived from a *linear* wave equation quite well, despite exhibiting vertical displacements $\chi$ that are not negligible when compared to the typical reduced height of $H_r \approx 3$ cm in the experiments (cf. Figs. 4 and 5). Generally, large interfacial amplitudes ($\chi \sim H_r$) are expected to necessitate considerable nonlinear corrections to the wave velocities, as demonstrated e.g. in a resonant two-sill configuration by Boschan et al. (2012). In the present setting, however, the linear wave theory performed fairly well. This finding is of relevance for actual sill-fjord dynamics, since there, too, pycnocline displacement $\chi$ is often comparable to $H_r$. In the case of the Gullmar fjord, for instance, internal amplitudes of $\chi \approx 5 - 10$ m are typical (Arneborg and Liljebladh, 2001), whereas the reduced height is $H_r \approx 19$ m (Parsmar and Stigebrandt, 1997). Thus our experiments confirm that it indeed appears to be sufficient to apply linear wave theory in such situations, as far as the wave speeds are concerned.

A further nondimensional quantity that is of relevance for the barotropic-baroclinic energy conversion is the ratio $\langle\chi\rangle/\langle\eta\rangle$ of the time-average vertical displacement at the pycnocline and at a surface antinode. The decay coefficient $C$ of the surface oscillation can also be measured using the natural timescale of the problem, the period of the dominant seiche mode as $C/\omega$. Expressing $C/\omega$ as a function of $\langle\chi\rangle/\langle\eta\rangle$ hence connects the ratio of the energy stored in the waves of the two layers (the energy scales with the respective squared amplitudes) and the rate of surface energy decay.

The result is shown in the log-log scatter plot of Fig.10 for the experiments in which the $\chi(t)$ record was available for evaluation. The data points appear to follow the empirical relationship $C/\omega \sim (\langle\chi\rangle/\langle\eta\rangle)^{0.28\pm0.04}$ (dashed line in Fig.10).

Taking the respective parameters from the Gullmar fjord field data as reported in Parsmar and Stigebrandt (1997) and Arneborg and Liljebladh (2001) yielded the red data point in Fig.10, whose error bars represent the range in which the data scatter. Interestingly, this data point also appears in the vicinity of the fitted curve, which suggests that the dynamics in the experimental and natural settings are indeed connected. The detailed investigation of this finding would, however, require field data from other sill fjords of similar bathymetry as well, which is beyond the scope of the present study.

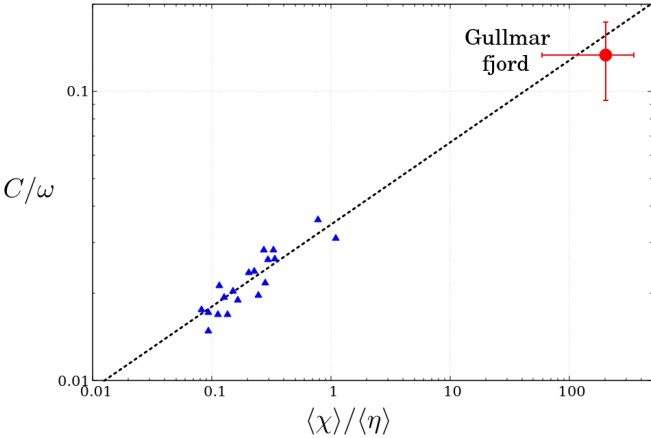

**Figure 10.** Nondimensional decay coefficients as a function of pycnocline vs. surface average displacement ratios for laboratory experiments (blue triangles) and for the Gullmar fjord (red circle). The dashed line represents the power law fit of $C/\omega = (0.035 \pm 0.002) \cdot (\langle\chi\rangle/\langle\eta\rangle)^{0.28 \pm 0.04}$

.

## 5 Conclusions

In this experimental study, we analyzed the coupling between surface seiche modes and internal wave dynamics in a quasi-two-layer stratified system, and the effect of a topographic obstacle on the damping of the surface seiche. The decaying surface oscillations were initiated by a wave maker in a pulse-like manner, and the adjustable shape of this device enabled us to excite surface standing waves of rarely observable larger wave-numbers.

Two pathways of barotropic-to-baroclinic conversion were uncovered. Firstly, the "direct" excitation of short-wavelength propagating internal waves via the horizontal velocity shear emerging in the vicinity of the obstacle. These waves were easily noticeable along the pycnocline due to their relatively large amplitudes. Their periods matched the dominant sloshing timescale associated with the surface seiche. It is to be noted that the most pronounced wave excitation, and thus the most prominent increase in the decay coefficient of the surface seiche was detected in the situations where the surface (standing) waveform was such that a node of the surface displacement was located above the bottom obstacle, that – in the geometry investigated here – means seiches of odd wave-number indices. (Only in these modes there is substantial current over the obstacle.)

Secondly, we found evidence for the excitation of slow, long wavelength internal seiche modes, whose half wavelength can fit roughly integer times onto the pycnocline between one sidewall and the obstacle (i.e. onto the half of the total basin length). The occurrence of such internal oscillations in this system is interesting as they have much lower characteristic frequencies than the surface seiche, i.e. the source signal. It appears, indeed, that a nontrivial source-filter dynamics can describe this phenomenon: the spectral structure of the decaying source signal includes non-zero amplitudes in the low-frequency range that

can resonate with certain internal standing wave modes whose wavelengths are such that they fulfil the geometrical boundary conditions, representing a "band-pass filtering".

Comparing the dominant seiche decay rates with those from control runs without bottom topography, we demonstrated that in almost all cases internal wave activity yields a detectable increase of the damping, also for even seiche mode indices, where the "short wave" excitation is inhibited, as discussed above. In these cases internal wave dynamics can be attributed to

the aforementioned slow wave (internal seiche) excitation pathway, but even this lesser effect appeared to be enough to yield substantially larger damping than the flat-bottom control runs.

Our findings are of relevance for the better understanding of baroclinic wave excitation in quasi two-layer sill fjords, as they demonstrate that the waves generated in such systems can be treated fairly well using the dispersion relations of the linear two-layer theory in cases where the surface forcing frequencies are large, e.g. in earthquake-induced fjord seiches reported

by Bondevik et al. (2013). We also demonstrated that in this particular system, nonlinear corrections to internal wave speeds are negligible, despite the fact that internal wave amplitudes are often comparable to the characteristic vertical scale of the stratification $H_r$, similarly to the case of natural sill fjords.

Future research is planned to address the source-filter feedback effects in the system to better understand the relationship between the decay rates of the waves at the surface and the ones along the pycnocline. and to explore the dynamics of a more

realistic, three-dimensional configuration, possibly involving the effect of Earth's rotation.

*Author contributions.* K.M. developed the set-up and performed the experiments. M.V., I.M.J. and K.M. conducted the evaluation of the data. All authors have contributed to the interpretation of results and the writing of the manuscript.

*Competing interests.* The authors declare that they have no conflict of interest.

*Acknowledgements.* We are thankful for the fruitful discussions with Tamás Tél and for the highly useful comments of editor John M. Huth-
nance and the three anonymous reviewers. This paper was supported by the National Research, Development and Innovation Office (NKFIH) under Grant FK125024. The work of K. M. is supported by the Stipendium Hungaricum Scholarship of the Tempus Public Foundation. This paper is also supported by the ÚNKP-18-4 New National Excellence Program (M.V.) of the Ministry of Human Capacities of Hungary. I.M.J. thanks the support by the Max Planck Institute for the Physics of Complex Systems in the framework of an Advanced Study Group on "Forecasting with Lyapunov Vectors".

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
