# Peer review of "Laboratory experiments on the influence of stratification and a bottom sill on seiche damping"

_Ocean Science, 2020_

## Referee Comment (RC1) · Anonymous Referee #1 · 12 Jan 2021

In this paper the authors set out to investigate, through a suite of lab experiment, the decay of internal wave motions generated at a topographic bump. The set up is interesting, and the introduction is very thorough in terms of geophysical importance and background for the work. A combination of theory and image analysis is used to quantify the wave properties and decay scales. The main conclusion of the paper is then that you get two IWs in the experiments set up here: one with the frequency of the forcing and one seiche, e.g., depending on the geometry of the basin. This is not quite a new result – it is how the ocean works. A quantification of the seiche period should be possible just from the stratification and tank geometry, but that is not done. Adding this would make the paper stronger and confirm that this is the mechanism. Also, is there a delay in the seiche generation compared to the immediate IW? That would shed light

[Figure]

Interactive
comment

on how the seiche is generated, e.g., if it draws from the surface motion or is set up by the IW. It is also shown that with an obstacle, there is a faster decay than with a flat bottom. The problem I'm having here is that there are not enough quantifications of a lot of the results mentioned in the text, and the physical mechanisms are only described briefly. More details are needed throughout on what is happening for a physical point. The main concern I have, however, is if this paper fits in Ocean Science. The exploration is very lab-based and theoretical, and while I really don't mind that, there are very few links to the real ocean. The concerns above requires substantial revisions linking the results to a geophysical setting to make the paper publishable in OS, but I actually think it may be better housed in a more theoretical fluid mechanics journal.

Minor comments L10: "in the bulk" -> rephrase; do you mean "water column"? L25: "...natural enclosed lakes, seas, fjords..." -> "...enclosed water bodies, e.g., lakes or fjords..." L91: "significantly different" -> this implies statistics have been done to show that they are indeed statistically different. Either show the stats, or rephrase so you don't use significant to mean large or substantial. Please do this throughout the manuscript. L94: "Whereas..." -> rewrite, this is an incomplete sentence. L100: I'm not sure what is novel about the result here: we know that we must have horizontal velocities over the obstacle to generate the internal waves – it is the pull and relaxation of the pycnocline over the obstacle that generates the waves. A description outlining this could be added to the text, and I think it explains the observations of IW propagation related to mode number discussed in the following paragraph as well? L116: "Such exemplary records..." -> "Examples of such records..." L128: "good agreement" – how good is good? Please quantify. L135: please comment further on the surprising result: why doesn't damping matter? Figure 2 has no legend relating the lines to the experiments. Figure 4: Please add panel labels (a, b, c...) and refer to them in the caption and the text.

---

## Referee Comment (RC2) · Anonymous Referee #2 · 22 Jan 2021

Review on " Laboratory experiments on the influence of stratification and a bottom sill on seiche damping"

by K. Meddjdoub, I.M. Janosi and M. Vincze

As explicitely said in the title of this article, this is an experimental laboratory study of surface wave excitation and damping in a two layer system. The measurement technique is quite simple and based on dye visualization, video recording and basic image analysis. If the introduction that presents the problem under study and the relevant bibliography is well done, I have not at all be persuaded by the scientific results. I do not recommend the publication of this article that does not contain enough material. Here are the main points that rise some problems:

[Figure]

1) The role of the obstable in the bottom layer is not clear and never studied. The flow around it should be measured and might explain its role in the interfacial waves generation.

2) The excitation by the wave maker is kind of obscure. Its motion should be qualified: what is its motion (amplitude, duration)? This could be done by video analysis.

3) Both interfaces motions should be analyzed through the space-time series recorded by the camera. The 2D FFT transform will thus show the experimental dispersion relations for each interface, to be compared with the classical surface wave theories.

4) The use of the Transfer function, simply defined by the ratio of the Fourier spectra of themotions of each interface is misleading: for instance, if the interfacial wave gains its energy at a given frequency by an other effect than linear direct energy transfer from the free surface mode, then the division by zero will make T(f) to diverge. I will recommend the use of cross-spectra that will show the energy exchanges between the Fourier modes.

5) The extraction of the energy damping coefficients is not explained but the results of major importance for the authors.

6) The authors claimed that the Fourier Transform of a damped sinusoïdal function possesses low frequency peaks. This is wrong in general. The Fourier Transform of exp(—at) sin(w0t) is : w0/[-wˆ2 + i w a + w0ˆ2 + aˆ2] and does not contain necessary low frequencies.

---

## Referee Comment (RC3) · Anonymous Referee #3 · 23 Jan 2021

The authors conduct a series of experiments in a flume with a two-layer fluid to study internal wave generation and propagation forced by a prescribed deformation of the free surface of the fluid, both with and without a topographic seamount. The authors report two paths of barotropic to baroclinic energy conversion.

My overall view is that a revised version of this paper might be suitable for publication in OS. In revising the paper I invite the authors to address the following points: 1. The experiments do not address the generation of seiches over the continental shelf, where the domain is "semi-infinite. The seiche generation mechanisms discussed rely on quantization of the wavelength along the axis of the tank, or between the obstacle and the ends of the tank. Clearly, if the domain is semi-infinite the seiche generation mechanisms will be modified. A discussion is required about this. Indeed, the authors

have overlooked the study by Davies, Xing and Willmott (2009) Ocean Dynamics, 9, 863. 2. The role of topography in seiche generation leaves for questions than answers. Why this shape of topography? Why is it always at a fixed point in the flume? From an oceanographic perspective it would be more interesting to have a representation of the continental shelf and slope. As it stands, the experiments discussed in this paper have at best tenuous relevance to the ocean. 3. The way the seiches are generated looks rather crude with the configuration of six foam bumpers. I am not convinced that you can accurately deform the free surface into the prescribed waveforms. Why not fabricate a solid material (planiform) with a surface that represents a linear external standing wave as characterised by the along channel modal number m? 4. The presentation of the results in the paper is sloppy. Please include a figure of the side elevation of the tank showing the two layer fluid, the depths H1, H_2, h, L, delta rho etc. The "golden rule" is that each mathematical symbol MUST be defined when it is first introduced in the paper. The authors appear to be unaware of this rule! 5. Why is there a problem with the m=5 standing wave? Using a more refined way of setting up the initial free surface displacement may well resolve this problem. 6. Figure 3, and elsewhere. A colour scale is required. 7. The analysis of time series of the interfaces was only conducted near the end walls of the flume. Why not at other locations. 8. The paper has not ben thoroughly proof read which is off putting for the referees. E.G. Line 10 "bulk" to "interior"; Figure 2 requires a definition of the symbols on each line; Table 1 the units of density are wrong; line 83 has a type; line 90, standing waves are not....;line 130, the dispersion relation has a typo; line after eqn (3), where k denote....; caption of figure 6, based on (6).... There is no eq (6)!

In conclusion, I would be willing to review revised version of this paper which addresses the above points. As it stands I am not able to recommend publication of this paper.

———————————————

---

## Editor Comment (EC1) · John M. Huthnance (Editor) · 11 Feb 2021

Dear Authors

You will have seen serious critical comments by three referees, relating to the description of what you have done and relevance to the ocean. One recommends rejection and the other two major revision; I would expect to send it back to at least two for re-review after revision. Therefore you will need to address these comments and include corresponding improvements in a revised manuscript before progress towards publication.

Yours sincerely

John Huthnance (editor)

---

## Author Comment (AC1) · 19 Mar 2021

We thank the referee for reading our manuscript and for the insightful and useful comments. Below we reply to the raised issues point by point.

Comment:

A quantification of the seiche period should be possible just from the stratification and tank geometry, but that is not done. Adding this would make the paper stronger and confirm that this is the mechanism.

Response:

The evidence supporting our statement that the observed frequencies of maximum

spectral amplification coincide with internal seiche mode frequencies is presented in Fig.6b of the manuscript in a way that is, to our understanding, equivalent to calculating the seiche frequencies (periods) that the referee suggests.

The internal wave dispersion relation curves shown in the plot are calculated from the parameters of the stratification only (using the theoretical formula given in eq. 3). The observation that the peaks of the measured maximum amplification (represented here with the color scale) occur at the integer values of non-dimensional wave number $L/\lambda$ (where L is the length of the tank) confirms the hypothesis that the amplification peaks are indeed coincide with frequencies associated with standing waves on the pycnocline i.e. internal seiche modes. In the revised version of the manuscript, we will make the presentation of this result clearer and emphasize its significance more.

Comment: Also, is there a delay in the seiche generation compared to the immediate IW? That would shed light on how the seiche is generated, e.g., if it draws from the surface motion or is set up by the IW.

Response:

Based on the referee's comment we have investigated this issue by taking the Wiener filtered time series of the pycncline displacement in the frequency bands of the surface oscillation and of the internal seiche mode at which the linear amplification (transfer function) was the largest. As an example, two typical filtered time series are presented in the attached Fig.1 of this rebuttal letter for the case of dominant surface wave modes m = 1 and m = 2 (experiment configuration #3). Unfortunately, however, our findings here were not entirely conclusive in terms of the delay between the two signals in any of the investigated cases.

It is to be noted, however, that marked low-frequency oscillations (and large transfer function peaks) were present in the cases of dominant surface wave modes m = 2 and m = 4 too (see Fig. 4 of the manuscript), where immediate IW generation at the obstacle is practically inhibited, as the dominant surface seiche then has an antinode

above the obstacle. This observations thus suggest that the presence of immediate IWs is not necessary for the excitation of the internal seiche modes.

Comment:

It is also shown that with an obstacle, there is a faster decay than with a flat bottom. The problem I'm having here is that there are not enough quantifications of a lot of the results mentioned in the text, and the physical mechanisms are only described briefly. More details are needed throughout on what is happening for a physical point.

Response:

Indeed, the main purpose of the present manuscript was to demonstrate the fact that the presence of a bottom obstacle reaching up to the pycnocline contributes significantly to the damping of the surface seiche. This may sound trivial, but to our surprise, we found that this simple mechanism was never actually investigated in laboratory experiments before, whereas various theoretical and field observational works have been dealing with the issue in very similar geometrical settings of the ocean system, e.g. fjords.

However, unfortunately (and unintentionally) we did not cite our paper published in Experiments in Fluids, dealing with a rather similar laboratory setting to the one studied here (Vincze and Bozóki 2017, temporarily made available for the reviewer at: http://karman3.elte.hu/mvincze/pub/13_exp_fluids_obstacle.pdf). In that work we extensively discussed the mechanism of internal wave generation above the obstacle and analyzed the velocity field using the technique of particle image velocimetry (PIV). There, however, we applied oscillatory forcing at the water surface, and therefore could not explore the dynamics of damping. Thus, in the present work our focus was indeed on this particular aspect, and we did not intend to duplicate our earlier work related to the wave generation. In the revised version of the manuscript, we will certainly add this reference and will summarize its key findings in the Introduction.
Comment:

The main concern I have, however, is if this paper fits in Ocean Science. The exploration is very lab-based and theoretical, and while I really don't mind that, there are very few links to the real ocean. The concerns above requires substantial revisions linking the results to a geophysical setting to make the paper publishable in OS, but I actually think it may be better housed in a more theoretical fluid mechanics journal.

Response:

Our exploration is lab-based and theoretical indeed, and now in retrospect we also agree with the referee that a fluid mechanics journal may have been a better fit (although we note that some papers discussing laboratory experiments have already been published in OS). However, the very reason we picked OS was the fact that this setting closely resembles the geometry of the ones studied by Stigebrandt and colleagues in their works related to barotropic-to-baroclinic energy transfer in fjords (e.g. the Gullmar fjord in Sweden) which we refer to in the introduction. These works investigate a semi-enclosed basin with seiche modes on the surface, a sharp density jump at the pycnocline and a topographic obstacle that reaches precisely up to the pycnocline, see the attached Fig.2 of this response letter, taken from Stigebrandt (1999).

We agree that neither this analogy with fjords nor the fact that the profiles are typical was emphasized enough in the previous version, hence we added a paragraph discussing these aspects to the Introduction section. To our understanding, fjords are a part of the ocean system, therefore we assumed – maybe incorrectly – that this experimental demonstration of the phenomenon may be of interest for the community. In the revised version of the paper we intend to emphasize these links more in the Introduction and the Discussion.

We also thank the referee for the minor comments: we agree with all of them and we will correct the text and figures accordingly.

[Figure]

**Fig. 1.**

[Figure]

Barotropic to
Baroclinic Energy
Transfer

1) $a_i \sin(\omega t - \varphi)$   tidal response

1) $a_o \sin(\omega t)$

2) $a_s = a_{so} \cdot e^{-c \cdot t}$  damped selche

4)

4)

3)

3)

**Fig. 2.**

---

## Author Comment (AC2) · 19 Mar 2021

We thank the referee for reading our manuscript and for making highly useful comments and suggestions even if recommending rejection. Below we reply to the raised issues point by point.

Comment:

1) The role of the obstacle in the bottom layer is not clear and never studied. The flow around it should be measured and might explain its role in the interfacial waves generation.

Response:
The present paper intended to focus on the damping effect due to barotropic-baroclinic energy conversion in a setting that has already been thoroughly investigated from the wave generation point of view in our earlier work, Vincze and Bozóki (2017) (available for the reviewer at: http://karman3.elte.hu/mvincze/pub/13_exp_fluids_obstacle.pdf). There we extensively discussed the mechanism of internal wave generation above the obstacle and analyzed the velocity field using the technique of particle image velocimetry (PIV). However, in that study, we applied oscillatory forcing at the water surface, and therefore could not explore the dynamics of damping.

It was our unintentional mistake that when listing earlier work in the introduction, we forgot to mention this paper, which is a predecessor of the present study. In the updated version of the manuscript we will definitely summarize our earlier findings related to the wave generation mechanism in the system.

Comment:

2) The excitation by the wave maker is kind of obscure. Its motion should be qualified: what is its motion (amplitude, duration)? This could be done by video analysis.

Response:

This was indeed done during our experiments. The characteristics of the vertical motion of the wave maker is demonstrated in Fig. 1. attached to the present response letter. (Based on the referee's comment, we will also incorporate this plot to the manuscript.) The dotted horizontal line (z = 0) represents the unperturbed water surface and the orange curve shows the vertical displacement of the runner foam "bumper" as a function of time. The width of the line represents the error of reproducibility. The total duration of the z < 0 interval was found to be (0.8 +/- 0.06) s and the amplitude (maximum depth w.r.t. the unperturbed water surface) was (3.65 +/- 0.25) cm.

Comment:

3) Both interfaces motions should be analyzed through the space-time series recorded by the camera. The 2D FFT transform will thus show the experimental dispersion relations for each interface, to be compared with the classical surface wave theories.

Response:

Based on the referee's comment we have carried out these analyses, and some exemplary plots will be included in the updated version of the manuscript. Fig.2 attached to the present response letter shows some of the results (both for the surface and for the internal interface in the selected cases of m = 4 and m = 6 dominant surface modes) with the theoretical surface and internal interface dispersion relations shown with solid curves.

Comment:

4) The use of the transfer function, simply defined by the ratio of the Fourier spectra of the motions of each interface is misleading: for instance, if the interfacial wave gains its energy at a given frequency by an other effect than linear direct energy transfer from the free surface mode, then the division by zero will make T(f) to diverge. I will recommend the use of cross-spectra that will show the energy exchanges between the Fourier modes.

Response:

The referee is of course right when stating that the transfer spectrum method can produce non-physical artifacts in case of nonlinear coupling between the source signal (surface wave) and the response signal (interfacial wave). Yet, such transfer functions are widely used tools for studying linear source-filter interactions in various fields (e.g. acoustics) and in our particular case it turned out that all of the significant peaks of the transfer function T(f) actually correspond to frequencies that are associated with internal seiche eigenmodes, as calculated from the dispersion relation (eq. 3).

For pairs of quasi-stationary signals cross power spectral density (CPSD) would indeed be a more informative way to present such results. Unfortunately though, here, probably due to the decaying nature of the time series in question, we were not able to acquire useful information using that method.

However, the wavelet spectrograms of the time series obtained using the so-called Morlet wavelet (aka Gabor wavelet) yielded reassuring results, when comparing their patterns to the T(f) transfer spectra. Wavelet transforms are generally more suitable to handle time series with time-dependent spectral structure (as the ones in this case). Some of our findings are presented in the attachment (Fig. 3) of this response letter, showing an exemplary T(f) transfer spectra (analogous to those presented in Fig.4 of the submitted manuscript) and the corresponding Wavelet spectrograms for the surface and interface time series extracted from the vicinity of the left-hand sidewall of the tank. Time scale ("period") tau, shown along the vertical axis represents the width of the Morlet window, and it gives higher wavelet coefficient values (color coding) when it fits locally to the time series. Thus, we can see the damping of surface modes (upper spectrogram), and the appearance and disappearance of various modes at the internal interface (bottom spectrogram).

On the bottom spectrogram, the notation is the following. The horizontal black dashed line is always the fundamental surface mode from the simple surface dispersion relation of Eq. (2). Dotted black lines belong to the peaks (simply tau = 1/f) in the transfer function. It is visible that the peaks of the transfer function T(f) indeed coincide with actual detectable oscillations. (Which, as it turns out in our Fig. 6b also coincide with internal seiche modes.)

Comment:

5) The extraction of the energy damping coefficients is not explained but the results of major importance for the authors.

Response:

Indeed, our explanation under equation (1) was not explicit, we will formulate it in a clearer manner in the updated version of the manuscript.

Taking the surface vertical displacement time series at the lateral sidewall of the rectangular tank (where it is assured, due to the boundary condition that all standing wave modes have antinode), we fitted the formula with N = 2 as a limit. It turned out (when checking the standard deviations of the residuals) that this two-term sum was sufficient to account for > 90% of the observed variance in all cases. This is how the frequencies and the damping coefficients of the most dominant modes (shown in Figs. 5 and 7) were acquired.

Comment:

6) The authors claimed that the Fourier Transform of a damped sinusoïdal function possesses low frequency peaks. This is wrong in general. The Fourier Transform of exp(âAËŸTat) sin(w0t) is : w0/[-wËȨ2 + i w a + w0ËȨ2 + aËȨ2] and does not contain necessary low frequencies.

Response:

We are well aware of the Fourier transform of the damping exponential function. The statement in question (line 142) has been the following: "even if the surface seiche was a perfectly 'monochromatic', single frequency source signal, its exponential decay would still unavoidably introduce nonzero amplitudes into the low-frequency range of its spectrum (see, e.g. French (1971)), making it suitable for the excitation of slow internal oscillations."

In the text did not intend to refer to "peaks" in the w < w0 range, merely stated that the spectral amplitudes are nonzero in this domain (as the spectral peak widens due to damping), which is true, also for the formula mentioned by the referee.

Also our statement in line 205 may very well have been misleading as it said: "the spectral structure of the decaying source signal includes low-frequency components

that can resonate with certain internal standing wave modes whose wavelengths are such that they fulfil the geometrical boundary conditions, representing a 'band-pass filtering'"

Thus, we reformulated the text, exchanging "low-frequency components" to "non-zero amplitudes in the low-frequency range" for clarity.

———————————

[Figure]

**Fig. 1.**

surface

interface

m = 4

m = 6

**Fig. 2.**

[Figure]

**Fig. 3.**

---

## Author Comment (AC3) · 19 Mar 2021

We thank the referee for reading our manuscript and for the insightful and useful comments and for stating that "a revised version of this paper might be suitable for publication in OS". Below we address the raised issues point by point.

Comment:

1. The experiments do not address the generation of seiches over the continental shelf, where the domain is "semi-infinite. The seiche generation mechanisms discussed rely on quantization of the wavelength along the axis of the tank, or between the obstacle and the ends of the tank. Clearly, if the domain is semi-infinite the seiche generation

mechanisms will be modified. A discussion is required about this. Indeed, the authors have overlooked the study by Davies, Xing and Willmott (2009) Ocean Dynamics, 9, 863.

Response:

We thank the reviewer for the suggested reference, that we will cite and briefly summarize in the updated version of the manuscript. As far as the surface seiche modes are concerned, the main difference between the closed and semi enclosed basins (i.e. gulfs) separated by a sill from the "semi infinite" open ocean is that in the latter case a node must be present at the obstacle (as also discussed by Davies, Xing and Willmott), similarly to the odd (i.e. antisymmetric) modes of our setting, in which the obstacle is situated in the midpoint of the tank. As for the internal seiche generation, a semi enclosed setting would have seiche modes between the end of the domain and the sill (as in our experiment) but no such standing waves could develop on the semi-infinite side. These differences indeed have to and will be mentioned when linking our setting to a natural setting of a gulf or fjord with an associated sill.

Comment:

2. The role of topography in seiche generation leaves for questions than answers. Why this shape of topography? Why is it always at a fixed point in the flume? From an oceanographic perspective it would be more interesting to have a representation of the continental shelf and slope. As it stands, the experiments discussed in this paper have at best tenuous relevance to the ocean.

Response:

We agree that this situation is not typical for the open ocean. However, this setting (a quasi-two layer stratification with an interface at sill depth) is often considered as a "quite good approximation of real fjord stratification because particularly dense water that occasionally refills the basin is trapped by the sill" (Stigebrandt, 1999), see also

the sketch attached to this response letter (Fig.1) taken from that reference.

We agree that neither this analogy with fjords nor the fact that the profiles are typical was emphasized enough in the previous version, hence we added a paragraph discussing these aspects to the Introduction section.

To our understanding, fjords are a part of the ocean system, therefore we assumed – maybe incorrectly – that this experimental demonstration of the phenomenon may be of interest for the community. In the revised version of the paper we intend to emphasize these links more in the Introduction and the Discussion.

Comment:

3. The way the seiches are generated looks rather crude with the configuration of six foam bumpers. I am not convinced that you can accurately deform the free surface into the prescribed waveforms. Why not fabricate a solid material (planiform) with a surface that represents a linear external standing wave as characterised by the along channel modal number m?

Response:

It is indeed true, that the surface cannot be deformed to prescribed waveforms with our excitation method, but our intention was not the excitation of pure modes. As indicated by eq. (1) the oscillation of the surface that actually develops in the system at the $x = 0$ location (left sidewall) is always a sum of various modes. Our goal was to acquire the natural frequencies and the frequency-dependent damping coefficients from the data by fitting the formula. Therefore, theoretically, any random initial surface shape could have provided the same information, since the Fourier components can be considered independent of each other in the linear approximation. Our "quasi pure" initiation method was applied because of practical reasons. As mentioned in the manuscript, even with this crude initiation a two-term sum ($N = 2$ in eq. 1), i.e. the combination of two eigenmodes was found to be sufficient to account for > 90% of the observed

variance in all cases, making the regression (the acquisition of eigenfrequencies and the corresponding damping coefficients) much easier. We will modify the text under eq. (1) accordingly, to emphasize this aspect in the updated version of the manuscript.

Comment:

4. The presentation of the results in the paper is sloppy. Please include a figure of the side elevation of the tank showing the two layer fluid, the depths H1, H_2, h, L, delta rho etc. The "golden rule" is that each mathematical symbol MUST be defined when it is first introduced in the paper. The authors appear to be unaware of this rule!

Response:

We thank the referee for the suggestion about the figure. We will add such a sketch to Fig.1 in the updated version of the manuscript. Although, it appears from the manuscript that, in accordance with the "golden rule" each symbol is defined in the text upon its first appearance (even if in a "sloppy" manner, for which we are sorry). However, these can still be rather inconvenient to find, we will therefore add a table to section 2 enlisting all the symbols that appear in the manuscript, alongside their definitions.

Comment:

5. Why is there a problem with the m=5 standing wave? Using a more refined way of setting up the initial free surface displacement may well resolve this problem.

Response:

Indeed, a surface oscillation with an m = 5 dominant standing wave mode could surely be excited with a more refined wave maker. However, as mentioned in our reply to comment 3, we did not intend to excite pure eigenmodes, and we could in fact observe m = 5 mode as a harmonic (albeit not as a dominant mode).

Comment:

6. Figure 3, and elsewhere. A colour scale is required.

Response:

A color scale has been added.

Comment:

7. The analysis of time series of the interfaces was only conducted near the end walls of the flume. Why not at other locations.

Response:

This position was selected to ensure that all standing wave modes have antinode – i.e. maximum amplitude – at the measurement location. Due to the boundary conditions the endwalls are the only such locations in the tank. A temporal spectrum taken here (and only here) has the same Fourier amplitudes as the total (space- and time-dependent) spectrum. Whereas (as an extreme case), at the midpoint of the tank, where all odd modes (m = 1,3,5,...) have nodes, one would get zero amplitudes for all these components in the local spectrum. Thus, the endwall appeared to be the best location for the analysis.

Comment:

8. The paper has not been thoroughly proof read which is off putting for the referees.

Response:

We are terribly sorry and we are very grateful for the referee for listing the mistakes. These (and more) will be corrected in the updated version of the manuscript.

[Figure]

**Fig. 1.**

---

## Author Response (AR2)

We are grateful for the Editor and the referees for re-reading our manuscript and for the insightful and highly useful comments. We also thank Referee #3 for recommending the acceptance of the previous revised version of the manuscript. Below we reply to the issues raised by the Editor and Referee #1.

**Comments from the Editor:**

*"Your laboratory values of parameters space do not tell the reader what may happen in a fjord that they are interested in. I do not think that you need to predict behaviour in all fjords, but you should say in what contexts your results could apply and in those cases what behaviour may be expected. Where do the results from Parsmar and Stigebrandt fit in with your results. "If the result were presented in a non-dimensional way, so they could be used by the ocean community" OS could be suitable. I wonder how the excess damping (obstacle compared with no obstacle) may depend on the amplitude(s) of the internal modes generated or their energy (input) relative to the surface wave. NB in the table of parameters, row for h, it should read (=H2)."*

**Comment from Referee #1:**

*"The issue here is that the authors admit that it isn't new that a sill dampens a seiche – that's what the paper cited shows – so what is then the point of showing that in the lab? The mechanism isn't new."*

*"...for example, where does the results from Parsmar and Stigebrandt fit in Figure 4? If the result were presented in a non-dimensional way, so they could be used by the ocean community, there may be mileage in the paper."*

**Combined response:**

Based on the above comments of the Editor and the Referee, we added an entire new section (Section 4, Discussion) in which we discuss the implications of our findings to natural seiche fjord systems, and compare our results to natural examples in terms of nondimensional quantities.

We indeed admit that certainly "it isn't new that a sill dampens a seiche" as the Referee put it, however, the fact that in this geometry the excited interfacial internal waves follow a linear dispersion relation (fairly well) despite their non-negligible amplitudes is probably of relevance for the community. The findings imply that in this particular setting nonlinear corrections to internal wave velocities are not necessary, even if the wave amplitudes are relatively large and the forcing period is small. We believe that this is important for the case of sill fjords with short seiching periods, and we hope that this result (as expressed in terms of nondimensional quantities) contributes to the better understanding of such systems. We tried to emphasize these connections in the new section.

Unfortunately, however, it was unclear to us how the Referee wished to connect the results of Parsmar and Stigebrandt to Fig.4 of the manuscript, which shows the space-time plots from the experiment, and primarily serves as a qualitative demonstration of the wave propagation in the system in case of two modes.

We also carried out an analysis comparing the interface-surface amplitude ratios to the nondimensional damping coefficients, as the Editor suggested, and discussed the possible relationship that may connect these ratios to the one observed in the case of the Gullmar fjord.

The typo discovered by the Editor has also been corrected.